# Emerging Anti-Inflammatory Pharmacotherapy and Cell-Based Therapy for Lymphedema

**DOI:** 10.3390/ijms23147614

**Published:** 2022-07-09

**Authors:** Ryohei Ogino, Tomoharu Yokooji, Maiko Hayashida, Shota Suda, Sho Yamakawa, Kenji Hayashida

**Affiliations:** 1Department of Frontier Science for Pharmacotherapy, Graduate School of Biomedical and Health Sciences, Hiroshima University, 1-2-3 Kasumi, Minami-ku, Hiroshima 734-8553, Japan; ryogino@hiroshima-u.ac.jp (R.O.); yokooji@hiroshima-u.ac.jp (T.Y.); 2Department of Psychiatry, Faculty of Medicine, Shimane University, 89-1 Enya-cho, Izumo 693-8501, Japan; maiko-s@med.shimane-u.ac.jp; 3Division of Plastic and Reconstructive Surgery, Faculty of Medicine, Shimane University, 89-1 Enya-cho, Izumo 693-8501, Japan; s.suda@med.shimane-u.ac.jp (S.S.); syama8@med.shimane-u.ac.jp (S.Y.)

**Keywords:** lymphedema, CD4+ T cell, mesenchymal stem/stromal cell, pharmacotherapy for lymphedema, regulatory T cell

## Abstract

Secondary lymphedema is a common complication of lymph node dissection or radiation therapy for cancer treatment. Conventional therapies such as compression sleeve therapy, complete decongestive physiotherapy, and surgical therapies decrease edema; however, they are not curative because they cannot modulate the pathophysiology of lymphedema. Recent advances reveal that the activation and accumulation of CD4+ T cells are key in the development of lymphedema. Based on this pathophysiology, the efficacy of pharmacotherapy (tacrolimus, anti-IL-4/IL-13 antibody, or fingolimod) and cell-based therapy for lymphedema has been demonstrated in animal models and pilot studies. In addition, mesenchymal stem/stromal cells (MSCs) have attracted attention as candidates for cell-based lymphedema therapy because they improve symptoms and decrease edema volume in the long term with no serious adverse effects in pilot studies. Furthermore, MSC transplantation promotes functional lymphatic regeneration and improves the microenvironment in animal models. In this review, we focus on inflammatory cells involved in the pathogenesis of lymphedema and discuss the efficacy and challenges of pharmacotherapy and cell-based therapies for lymphedema.

## 1. Introduction

Lymphedema is caused by a dysfunction of the lymphatic system, resulting in localized interstitial fluid retention and tissue swelling; it is classified as primary or secondary. Primary lymphedema develops due to inherited hypoplasia/dysplasia or dysfunction of lymphatic vessels because of some intrinsic factors such as genetic mutations in the signaling pathway for vascular endothelial growth factor C (VEGF-C), while secondary lymphedema is caused by a dysfunction of the lymphatic vascular system due to trauma or parasitic infection [1,2,3]. Although the incidence of secondary lymphedema has been declining due to advances in surgery, this iatrogenic disorder has a strong negative impact on physical and mental quality of life (QOL) [4]. Additionally, radiation therapy (RT) increases the risk of lymphedema in the upper and lower limbs; for instance, in breast cancer patients, the risk of lymphedema is five times higher with postoperative RT than with axillary lymph node dissection alone [5]. Patients with lymphedema typically present with symptoms such as altered mechanical properties and sensitivity of the skin, increased susceptibility to systemic and local infections, decreased function of the affected upper or lower limb, and chronic pain and discomfort [3,6]. In addition, patients may have problems with body image and social acceptability and exhibit low self-esteem [6]. The protein-rich fluid accumulated in the interstitial space induces the migration of CD4+ T-helper (Th) cells, low-grade inflammation, remodeling of extracellular matrix, hyperkeratosis, adipose deposition, and fibrosis [6,7,8,9,10,11]. These changes in the edematous limb exacerbate lymphatic dysfunction, resulting in clinical manifestations of lymphedema.

No curative therapy for lymphedema has been established so far. While conservative therapies (such as manual lymphatic drainage, complete decongestive physiotherapy, compression sleeve therapy, exercise, and weight reduction) decrease edema temporarily, they cannot modulate the pathophysiology of lymphedema. Therefore, it is difficult to maintain their therapeutic efficacy over a lifetime [12,13,14,15,16,17]. Surgical interventions such as lymphovenous anastomosis/bypass or vascularized lymph node transfer (VLNT) are effective in early-stage lymphedema; however, they are ineffective in chronic lymphedema with fibrosis due to lymphatic dysfunction in the edematous region [18,19,20,21,22]. Recently, pharmacotherapy and cell-based therapy have been developed to treat lymphedema by promoting lymphangiogenesis, improving lymphatic function, and suppressing fibrosis and inflammatory responses. Several studies focus on the migration and accumulation of CD4+ T cells in the edematous region as a new target to treat lymphedema [23,24,25,26,27,28]. Mesenchymal stem/stromal cells (MSCs) exert anti-inflammatory, anti-fibrosis, antioxidant stress, and immunomodulatory effects and are hence used in studies to establish cell-based therapy to treat wounds [29], inflammatory bowel diseases [30], diabetes mellitus [31], psoriasis [32,33], and graft-versus-host disease [34]; they are useful since they promote lymphangiogenesis in lymphedema animal models [35,36,37,38,39,40,41,42,43,44,45]. In this review, we focused on the inflammatory cells involved in the pathogenesis of lymphedema and discuss the efficacy and challenges of pharmacotherapy and cell-based therapies for lymphedema.

## 2. Pathophysiology of Secondary Lymphedema

In most regions of the body, lymph flows against a hydrostatic pressure gradient created by extrinsic and intrinsic pump forces arising from the surrounding skeletal muscle and/or lymphatic collecting vessel network [46]. In lymphedema, fluid stasis is caused by lymphatic pump dysfunction due to contractile dysfunction, chronic inflammation, fibrosis, abnormal lymphangiogenesis, barrier dysfunction, or valve defects [46,47]. The protein-rich fluid that accumulates triggers an inflammatory response and exacerbates lymphedema. Although details of the mechanisms remain unclear, the activation of dendritic cells (DCs) and subsequent activation of CD4+ T cells, especially Th2 cell maturation, have been hypothesized as key factors in the inflammatory response (Figure 1) [11,48].

In the early phase (~6 weeks) of lymphatic injury, endogenous danger signals such as high mobility group box 1 (HMGB1) and heat-shock protein 70 are expressed in endothelial cells, adipocytes, and other stromal cells at the injury site [11,26]. These proteins promote lymphangiogenesis via toll-like receptor signaling and the blockade of HMGB1 activity with glycyrrhizin inhibited inflammatory lymphangiogenesis in the mouse tail lymphedema model [11,26,49,50]. Furthermore, macrophages are recruited, and they accumulate in the lymphedematous region, especially in the early phase [51,52], while M2 macrophages secrete VEGF-C to promote superficial lymphangiogenesis [53]. Shimizu et al. reported that bone marrow-derived M2 macrophages may serve as lymphatic endothelial cell (LEC)-progenitors after adipose-derived regenerative cell (ADRC) treatment in mouse tail lymphedema models [38]. Therefore, innate immune responses may promote lymphangiogenesis and suppress the development of lymphedema in the early phase of lymphatic injury.

In this early phase, DCs accumulate in the injured skin and activate acquired immunity [25]. The expression of C–C chemokine receptor (CCR) type 7 increases on the surface of activated DCs, which migrate according to the concentration gradient of C-C chemokine ligand (CCL) type 21 secreted by LECs. Furthermore, after reaching the lymphatic vessels, activated DCs invade them using intercellular adhesion molecule 1 and/or vascular cell adhesion molecule 1 and flow into the draining lymph nodes [54,55,56,57,58]. In lymph nodes, naïve CD4+ T cells are activated by DCs, and increase the expression of cutaneous leukocyte antigen (CLA), CCR4, CCR9, and CCR10. Furthermore, after entering the bloodstream, activated CD4+ T cells infiltrate the edematous region using adhesion molecules such as E-selectin (a CLA ligand) and migrate toward chemokine ligands for CCR4 (CCL17) and CCR 10 (CCL27). The expression of these adhesion molecules and chemokines increases in the vasculature and keratinocytes of lymphedematous tissue, respectively [25,59,60,61,62,63]. Subsequently, inflammatory cytokines such as interferon (IFN)-γ, interleukin (IL)-4, IL-13, and transforming growth factor (TGF)-β (secreted from the activated CD4+ T cells) promote infiltration of the inflammatory cells, exacerbate fibrosis by collagen deposition, and directly inhibit lymphangiogenesis by suppressing the proliferation, differentiation, and migration of LECs [10,24,64,65,66,67,68,69,70]. Therefore, the activation of CD4+ T cells through antigen presentation by DCs is the key process in the development and exacerbation of lymphedema. This hypothesis is supported by studies using the mouse tail lymphedema models, in which CD4 knockout mice were less likely to develop lymphedema [25] and the depletion of regulatory T cells (Tregs) exacerbated lymphedema [23].

M1 macrophages also infiltrate the lymphedematous region and exacerbate lymphedema by inducing adipose deposition and chronic inflammation via IL-6 [71,72]. In addition, M1 macrophages strongly express inducible nitric oxide synthase (iNOS) and disturb nitric oxide homeostasis maintained by endothelial nitric oxide synthase (eNOS), resulting in attenuated lymphatic vessel pumping [25,73]. These reports suggest that the innate immune system, especially M1 macrophages, is involved in aggravating lymphedema as an inflammatory reaction in the chronic phase.

## 3. Pharmacotherapy for Lymphedema

Doxycycline, ketoprofen, ubenimex, selenium, synbiotic supplement, tacrolimus, anti-IL-4/IL-13 antibody, fingolimod, and TGF-β inhibitors have been studied for their suppression of inflammatory and oxidative stress. The reported pharmacological mechanisms of these agents are summarized in Figure 2.

### 3.1. Doxycycline

Doxycycline, a tetracycline antibiotic, is an anti-Wolbachia drug used for filarial lymphedema. Mand et al. reported that a 6-week course of doxycycline at 200 mg/day improved mild to moderate lymphedema for two years, independent of ongoing filarial infection [74]. The efficacy of doxycycline is probably due to its non-antibiotic effects such as the direct inhibition of inflammation and angiogenesis [75].

### 3.2. Leukotriene B4 Inhibitors (Ketoprofen, Ubenimex)

Ketoprofen is a non-steroidal anti-inflammatory drug. The efficacy of ketoprofen has been demonstrated in a mouse tail lymphedema model and patients with lymphedema. In a mouse lymphedema model, the subcutaneous injection of ketoprofen decreased tail volume and suppressed histological changes such as epidermal thickening and neutrophil infiltration, while increasing the expression of tumor necrosis factor-α (TNF-α). In contrast, pegsunercept, a modified soluble form of TNF-α receptor R1, increased tail volume, histologically exacerbated the disease, and reduced TNF-α expression. The expression of VEGF-C in this model showed a correlation with TNF-α expression, suggesting that ketoprofen could induce TNF-α-dependent VEGF-C expression followed by lymphangiogenesis [76]. In a clinical study, patients with lymphedema received 75 mg oral ketoprofen three times daily for four months [77]. This treatment significantly improved the histopathology scores (dermal thickness, collagen thickness, intercellular mucin deposits, and perivascular inflammation); however, the volume of the limbs and content of the extracellular fluid were not affected. The mechanism of action of ketoprofen in lymphedema is the inhibition of 5-lipoxygenase activity, which produces leukotriene B4 (LTB4), rather than the inhibition of cyclooxygenase activity [77,78]. In a mouse tail lymphedema model, the intraperitoneal injection of ubenimex (2 mg/kg), a leukotriene A4 hydrolase inhibitor, improved lymphatic collecting vessel pumping, but did not affect the tail volume and leukocyte population in draining lymph nodes [79]. However, the long-term administration of ketoprofen may be inappropriate because of side effects such as acute kidney injury and gastric ulcer due to the non-selective inhibition of physiological cyclooxygenase activity. In contrast, ubenimex inhibits the production of LTB4 selectively. Therefore, ubenimex may be more suitable than ketoprofen for long-term treatment.

### 3.3. Selenium

The oral and intravenous administration of sodium selenite is effective in the treatment of lymphedema associated with breast cancer and head and neck cancer. In these patients, selenium decreased edema volume and improved the clinical stage of lymphedema. The antioxidant properties of selenium contribute to its efficacy in lymphedema; however, the experimental evidence is unclear [80,81,82]. In a recent study, the intravenous administration of sodium selenite improved lymphoedema and elevated the serum levels of corticosterone, LTB4 dimethylamide (endogenous LTB4 antagonist), and prostaglandin E3 in breast cancer-related lymphedema patients [83]. Elevated levels of these anti-inflammatory substances may be a factor in the therapeutic efficacy of selenium.

### 3.4. Synbiotic Supplements

Synbiotic supplements, dietary supplements combining probiotics and prebiotics, reduce inflammatory markers such as C-reactive protein (CRP) and TNF-α [84]. In overweight and obese patients with breast cancer-related lymphedema, a 10-week combination of low-calorie diet and synbiotic supplementation resulted in significant reductions in edema volume, serum leptin, and serum inflammatory marker levels (high-sensitivity CRP, IL-1β and TNF-α). However, after adjusting for baseline edema volume, inflammatory marker levels, and body mass index, only serum leptin and TNF-α levels were found to be significantly lower in the synbiotic supplementation group than in the low-calorie diet and placebo capsule group [85]. Although the same research group has reported antioxidant effects and improvement in QOL score with the same dietary combination, the efficacy of synbiotic supplementation for lymphedema has not yet been determined [86,87].

### 3.5. CD4+ T Cell Suppressants (Tacrolimus, Anti-IL-4/IL-13 Antibodies, Fingolimod)

Treatment with tacrolimus ointment and anti-IL-4/IL-13 antibodies has been shown to suppress the activation and differentiation of CD4+ T cells [24,27,28]. Tacrolimus ointment is used to treat cutaneous inflammatory diseases such as atopic dermatitis. In CD4+ T cells, tacrolimus binds to FK-506-binding protein 12, and the complex inhibits the phosphatase activity of calcineurin, thereby reducing the transcription of IL-2 [88]. CD4+ T cells cannot survive in the presence of tacrolimus because optimal autocrine IL-2 signaling is essential to limit the apoptosis of effector CD4+ T cells and to sustain their transition to and persistence as memory cells [89]. The local administration of tacrolimus in a mouse model with tail lymphedema showed protective and therapeutic efficacy by reducing soft tissue thickness, suppressing inflammatory cell infiltration and inflammatory cytokine expression, and increasing the formation of lymphatic-collecting vessels at the injured site. Furthermore, the recovery of lymphatic functions by tacrolimus, including lymphatic pumping, was observed in a popliteal lymph node dissection model [24].

The inhibition of Th2 differentiation with IL-4- or IL-13-neutralizing antibodies prevents the initiation and progression of lymphedema by inhibiting tissue fibrosis and improving lymphatic function in another mouse tail lymphedema model [27]. Additionally, a report detailed the efficacy of monthly intravenous QBX258 infusion, a combination of two monoclonal antibodies neutralizing IL-4 and IL-13, in eight patients with breast cancer-related lymphedema [28]. Four infusions of QBX258 reduced histological epidermal thickness and suppressed keratinocyte proliferation, type III collagen deposition, mast cell infiltration, and Th2-inducible epithelial-derived cytokine (IL-33, IL-25, and thymic stromal lymphopoietin) expression. QBX258 also improved skin stiffness and patient QOL scores immediately after treatment; however, these improvements returned to baseline four months after the treatment was discontinued. Furthermore, treatment with QBX258 did not decrease the limb volume [28].

Fingolimod (FTY720), a modulator of the sphingosine-1-phosphate receptor, suppresses the emigration of lymphocytes from lymph nodes. In a mouse model with popliteal lymph node dissection, the administration of fingolimod (dissolved in drinking water) from the day of surgery increased CD4+ T cells in inguinal lymph nodes, but decreased them in the skin of the hindlimb [25]. In another mouse tail lymphedema model, fingolimod suppressed the increase in edema volume and fibroadipose thickness from 1 to 6 weeks after lymphatic dissection [25].

### 3.6. TGF-β Inhibitors (Anti-TGF-β Antibody, Vactosertib, LY-364947)

TGF-β is one of the key mediators in tissue fibrosis; it inhibits functional lymphatic regeneration in the lymphedematous region. The inhibition of TGF-β signaling by monoclonal antibodies or small-molecule drug EW-7197 (vactosertib, an inhibitor of TGF-β receptor type 1) enhanced lymphangiogenesis and lymphatic function by inhibiting fibrosis in a mouse tail lymphedema model [10,90]. Furthermore, LY-364947, a selective inhibitor of TGF-β receptor type 1, markedly suppressed tissue fibrosis and improved lymphatic dysfunction, which was induced by irradiation (with 15 Gy radiation) on the mouse tail [91].

## 4. Cell-Based Therapy for Lymphedema

### 4.1. Animal Studies

Cell-based therapies have been studied using lymphedema animal models, as shown in Table 1. In these studies, MSCs derived from bone marrow or adipose tissue were commonly used; until around 2010, the primary mechanism of cell-based therapy using MSCs was attributed to their multipotency in differentiating directly into LECs and promoting lymphangiogenesis [35,36,37,43]. Hwang et al. constructed a mouse hindlimb lymphedema model by circumferential incision and electrocautery of the lymph vessels in the thigh. On the day of surgery, PKH-26-labeled human adipose-derived MSCs (ASCs) were injected subcutaneously, and VEGF-C hydrogel sheets were sutured to the site of the injured lymphatic vessels. As a result, the footpad thickness of the affected limbs was significantly reduced, and the number of vessels with lymphatic vessel endothelial hyaluronan receptor 1 (LYVE-1) was significantly increased on day 28 after the operation. Furthermore, the co-localization of LYVE-1 and PKH-26 was observed around the lymphatic vessels in the combination of human ASCs and VEGF-C hydrogel, but not in human ASCs alone, indicating that transplanted ASCs combined with VFGF-C hydrogel could differentiate into LECs in vivo under specific conditions [36]. In addition, Dai et al. established a mouse hindlimb lymphedema model by circumferentially incising the thigh and removing inguinal lymph nodes following two rounds of 2.25 Gy radiation. Six weeks after the surgery, podoplanin-positive ASCs derived from green fluorescent protein (GFP)-transgenic mice were injected into the lymphedematous skin. Two weeks after implantation, co-localization of GFP and LYVE-1 was detected only in the lymphatic vessels of the podoplanin-positive ASCs transplantation group, and not in the podoplanin-negative and unsorted ASCs transplantation groups [43]. However, the ASCs used in the study may be adipose-derived stromal vascular fraction (SVF) and the cell population of their podoplanin-positive ASCs may be LEC-progenitor cells contained in the SVF.

However, the engraftment and direct differentiation of MSCs into LECs may only be marginally effective. The differentiation of ASCs into LECs after transplantation has not been confirmed because SVF or ASCs derived from GFP-transgenic mice cannot survive long term [38,40,43]. Recent studies have focused on paracrine functions that exert anti- inflammatory, anti-fibrosis, and immunomodulatory effects. In animal studies, the therapeutic effects of MSCs increased the lymphatic vessel density (by secreting lymphangiogenic factors such as VEGF-C) [35,36,37,38,39,40,41,42,43,44], restored lymphatic vessel function (by promoting the regeneration of lymphatic collecting vessels and lymphatic pumping) [35,41,42,45], promoted wound healing [39,42], and improved the tissue microenvironment (by anti-fibrotic and anti-inflammatory effects) [38,39,44].

MSC-based therapies combined with biomaterial scaffolds such as Matrigel^®^ [42] and BioBridge^®^ [45] or surgical therapy (VLNT) [41] exhibit efficient regeneration of the functional lymphatic system. These combinational therapies are expected to have a more synergistic effect on lymphedema than individual therapy.

In addition to MSCs, LECs and Tregs have been considered candidate cell populations for lymphedema therapy [23,92,93]. Both LEC-like cells differentiated from mouse muscle-derived stem cells and human LECs promote functional lymphangiogenesis by lymphography [92,93]. Although human LECs did not survive in F344/N rnu/rnu nude rats for more than 22 days after local injection, this treatment resulted in reduced skin thickness and the regeneration of rat-derived lymph vessels. These results suggest that transplanted LECs not only integrate in regenerated lymphatic vessels, but also promote the secretory function of resident cells [93]. Gousopoulos et al. evaluated the efficacy of Tregs (CD4+, CD25+ T cells) transplantation (intravenous injection) using a mouse tail lymphedema model. The results showed the suppression of tail volume increase, inflammatory cell (CD45+ cells, CD206+ cells, and CD68+ cells) infiltration, Tgfb1, Tnfa, and Il10 mRNA expression and fibrotic tissue deposition, and restored lymphatic vessel dilation and lymphatic flow. Therefore, the transplantation of Treg that suppresses Th1/Th2 immune responses has the potential to be a novel therapeutic strategy [23]. The systemic expansion of Tregs by intraperitoneal injection with IL-2/anti-IL-2 monoclonal antibody complexes (IL2-c) was effective for lymphedema to the same degree as adoptive Treg transplantation [23]. However, IL2-c therapy may not be clinically suitable due to the potential side effects of effector T cell activation. Although it has not been confirmed whether MSCs transplantation could induce Tregs in lymphedema models, the expansion of Tregs by MSCs has been examined in in vitro and in vivo studies [94,95]. Therefore, the induction of Tregs may contribute to the therapeutic efficacy of MSCs transplantation in lymphedema.

### 4.2. Clinical Studies

Somatic stem cell-based therapy for lymphedema in humans has been reported by five research groups (Table 2). The therapeutic efficacy of autologous transplantation was evaluated with bone marrow-derived MSCs or ADRCs for patients with breast cancer-related lymphedema [97,98,99,100,101] and with bone marrow-derived mononuclear cells and peripheral blood hematopoietic stem cells for patients with primary lower limb lymphedema [102,103]. The volume of the edematous limb decreased with cell-based therapy in four groups, while subjective symptoms including heaviness, tension, pain, sensitivity, and mobility of the affected limb and total QOL scores improved in all groups; these effects continued throughout the 4-year follow-up [101]. At the 4-year follow-up after ADRC transplantation, six of the ten patients had reduced the use of conservative lymphedema therapy, and no serious adverse events were observed in the patients [101]. While minor adverse events such as bruising, pain, itching, reduced sensation, and slight irregularity of the skin surface were observed at the donor site, these resolved spontaneously up to 6 months after transplantation [99]. Thus, somatic stem cell-based therapy is a new therapeutic strategy for improving subjective and objective symptoms of lymphedema and for decreasing the bothersome conservative therapy in the long term. Further analyses underlying the therapeutic mechanisms in clinical settings are desirable.

## 5. Discussion

Anti-inflammatory pharmacotherapy and cell-based therapy are new therapeutic strategies for improving lymphedema symptoms by promoting functional lymphangiogenesis and improving the microenvironment of the edematous region. However, there has been little evidence for their usefulness in lymphedema therapy. Some animal studies and clinical pilot studies have shown that these therapies could improve the subjective symptoms, but not the volume or appearance of lymphedema. In this review, we focus on the inflammatory cells involved in the pathogenesis of lymphedema, and discuss the clinical usefulness of pharmacotherapy and cell-based therapies for lymphedema.

Since it is difficult to reflect the pathophysiology of secondary lymphedema in animal models because of its occurrence in the chronic state, many animal models have been established to evaluate the new therapeutic strategy for lymphedema. The rodent tail lymphedema model is commonly used to evaluate the therapeutic strategies for secondary lymphedema because this model closely mimics the progression of human lymphedema, including fibrosis, fat deposition, and the infiltration of immune cells [104]. However, it is questionable whether rodent tail lymphedema mimics the pathophysiology of lymphedema in human limbs because the tail of rodents has no lymph nodes and is anatomically and physiologically different from human limbs [104,105]. The mouse hindlimb lymphedema model is also used as an animal model of lymphedema. This model is created by a combination of surgery and irradiation, and closely represents the chronic lymphedematous state in humans. When lymphedema is induced by surgery alone, the edema might resolve spontaneously. Thus, irradiation is often necessary to create the chronic lymphedema model. However, the dose and timing of irradiation are not standardized, and the degree of edema varies widely among studies [104]. Although animal lymphedema models such as rabbit, sheep, dog, pig, and monkey have also been used to study chronic or clinically-relevant lymphedema, the numbers of these studies are limited [104]. Therefore, the establishment of a common animal model that reflects the pathophysiology of human lymphedema is necessary to develop new therapeutic strategies.

Doxycycline, selenium, and synbiotic supplements have been reported to improve lymphedema symptoms to some extent. However, their efficacies for decreasing edema volume and improving the clinical stage of lymphedema are not remarkable. Furthermore, their therapeutic mechanisms for lymphedema have not been elucidated. Compared to these agents, the therapeutic mechanisms of immunosuppressive agents such as tacrolimus, IL-4/IL-13-neutralizing antibodies, and fingolimod have been elucidated. Although these immunosuppressive agents improve the lymphedema symptoms, the symptoms may return to baseline after the treatment is discontinued. Therefore, patients with lymphedema require lifelong treatment to maintain their QOL. However, long-term treatment with immunosuppressive agents such as fingolimod and IL-4/IL-13-neutralizing antibodies may be a risk factor for infection (e.g., cellulitis) [28]. Further evidence of the long-term safety and duration of immunosuppressive therapies is necessary to treat lymphedema in clinical settings. Collectively, we consider that pharmacotherapies with these agents cannot be recommended actively for lymphedema at present.

MSCs exhibit immunomodulatory effects [106], and Jørgensen et al. suggested that ADRC transplantation could alleviate the incidence of cellulitis in patients with breast cancer-related lymphedema during a 4-year follow-up period [101]. Additionally, bone marrow-derived mononuclear cells and peripheral blood hematopoietic stem cells could decrease edema volume in primary lower limb lymphedema without serious adverse events [102,103]. Although the molecular mechanism underlying their therapeutic efficacy is unclear at present, cell-based therapy is attractive to establish new therapeutic strategies for primary and secondary lymphedema. We speculate that transplanted MSCs may improve lymphedema symptoms through anti-inflammatory, antioxidant, and immunomodulatory effects via cytokines/growth factor secretion, Tregs induction, and improvement in the microenvironment.

A major problem in current cell-based therapy is that MSCs are hardly characterized by cell surface markers and the multipotency of the isolated cell population. In 2006, the International Society for Cell & Gene Therapy Mesenchymal Stromal Cell committee proposed that the minimal criteria defining human multipotent mesenchymal stromal cells, rather than mesenchymal stem cells, were plastic adherence, expression of (≥95% positive) CD105, CD73, and CD90, lack of (≤2% positive) hematopoietic and endothelial markers CD45, CD34, CD14 or CD11b, CD79α or CD19, and HLA-DR, and the capability of differentiation into adipocyte, chondrocyte, and osteoblast lineages in vitro [107,108]. However, most MSC populations used for lymphedema therapy have not been characterized by flow cytometry or differentiation capacity assay. Furthermore, cell surface markers for animal-derived MSCs have not been defined. Since adipose-derived MSCs, especially fresh SVF or ADRCs, belong to a heterogeneous cell population, the therapeutic efficacy of ASC transplantation may vary among researchers and/or physicians. Additionally, Bucan et al. reported no differences in the hindlimb volume and lymphatic clearance of a hindlimb lymphedema mouse model between the SVF or ASC transplantation and vehicle control groups. In their study, <20% of ASCs expressed CD105 and/or stem cell antigen-1 [96]. Therefore, the characterization of MSCs used in therapy can help to eliminate the differences in therapeutic efficacy among practitioners.

In the future, the treatment of lymphedema should focus on lymphatic regeneration and on improving the microenvironment of the edematous region, such as the suppression of fibrosis and infiltration of inflammatory cells, and the regulation of CD4+ T cell balance. Although further evidence on the long-term safety and efficacy and underlying mechanisms of anti-inflammatory pharmacotherapy and cell-based therapy is necessary, these therapies may shed light on the development of a new radical therapeutic strategy for lymphedema to improve the microenvironment and immune responses.

## Figures and Tables

**Figure 1 ijms-23-07614-f001:**
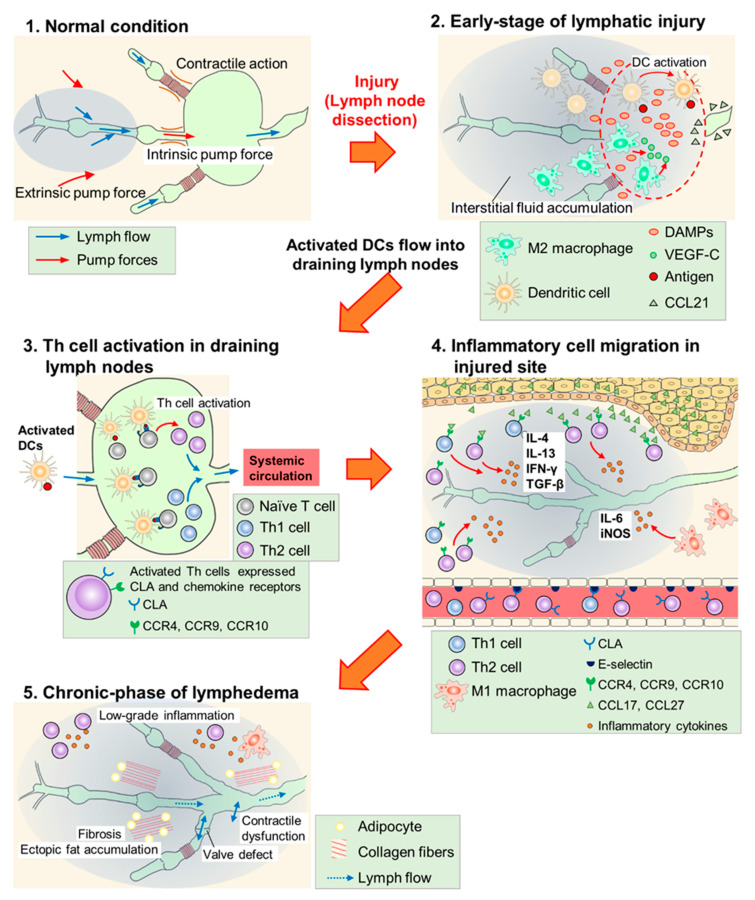
Scheme of lymphedema development after lymph node dissection. (**1**) In normal conditions, lymph flow is generated due to intrinsic pump force by a lymphatic collecting vessel network and extrinsic pump force by surrounding skeletal muscles. (**2**) In the early phase of lymphatic injury, damage-associated molecular patterns (DAMPs) are released from injured cells and these molecules promote lymphangiogenesis. M2 macrophages secrete vascular endothelial growth factor C (VEGF-C) and serve as lymphatic endothelial cell (LEC) progenitors. Dendritic cells (DCs) are activated at the injured site, and invade into lymphatic vessels along the concentration gradient of the C–C chemokine ligand (CCL) 21 secreted by LECs. (**3**) Activated DCs flow into draining lymph nodes and activate helper T (Th) cells. Expressions of cutaneous leukocyte antigen (CLA), C–C chemokine receptor (CCR) 4, CCR9, and CCR10 are increased at the surface of activated Th cells. These cells enter systemic circulation. (**4**) Activated Th cells, guided by adhesion molecules and CCLs, infiltrate the injured site and secrete inflammatory cytokines. M1 macrophages also accumulate at the injured site and cause inflammatory responses. (**5**) Low-grade inflammatory responses, fibrosis, adipose deposition, and unfunctional lymphangiogenesis (valve defect and contractile dysfunction) occur in the chronic phase of lymphedema. These responses impair lymphatic function and exacerbate lymphedema.

**Figure 2 ijms-23-07614-f002:**
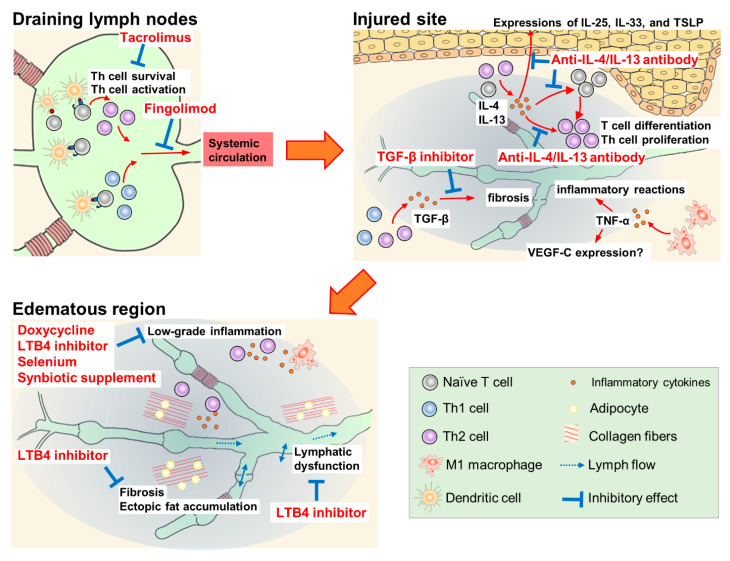
Pharmacological mechanisms of therapeutic agents for lymphedema.

**Table 1 ijms-23-07614-t001:** Cell-based therapies in animal models of lymphedema.

Author	Animal	Model of Lymphedema	Cell	Treatment	Control Group(s)	Outcomes
(Transplantation of MSCs)						
Conrad et al. 2009 [35]	Female C57BL/6 mouse	Tail model Surgery alone	p53-/- mice origin BMSC	1 × 10^7^ cells/animal Local subcutaneous injection Once a week injection (timing of the first injection is uncertain)	Non-treatment	Decreased circumference in edematous region (POD 27) Restored lymphatic drainage across the site of incision (POD 56) Increased the number of LYVE-1- and podoplanin-positive lymphatic vessels (POD 56)
Hwang et al. 2011 [36]	Female BALB/c mouse	Hindlimb model Surgery alone	Human origin ASC, PKH-26-labeled (commercial item)	Injected cell numbers not described Local subcutaneous injection + VEGF-C hydrogel sheet suture at POD 0	Non-surgery Non-treatment ASC alone VEGF-C hydrogel sheet alone	Decreased footpad thickness of hindlimb (POD21) Increased lymphatic vessel density (POD 28) Detected the co-localization of PKH-26 and LYVE-1 in injured site (POD 28)
Zhou et al. 2011 [37]	Female/ male New Zealand white rabbit	Hindlimb model Surgery + RT ^60^Co γ-ray irradiation, 2000 cGy, 3 days after surgery	New Zealand white rabbit origin BMSCs (CD29+, CD44+, CD11b−, CD45−)	1 × 10^7^ cells/animal + VEGF-C 150 ng/kg Local intramuscular injection at ~3 months after operation	Vehicle alone BMSC alone VEGF-C alone	Decreased edematous limb volume in BMSC alone, and VEGF-C alone group, and further decreased in BMSC + VEGF-C group (28 days and 6 months after treatment) Increased lymphatic vessel numbers in BMSC alone, and VEGF-C alone group, and further increased in BMSC + VEGF-C group (28 days after treatment) Increased protein expression of VEGF-C at transplantation areas in BMSC + VEGF-C group and BMSC alone groups
Shimizu et al. 2012 [38]	Male C57BL/6J mouse	Tail model Surgery alone	Mouse inguinal fat pad origin ADRC	2 × 10^6^ cells/animal Local subcutaneous injection in 2 different points at POD 1	Sham control Vehicle alone	Decreased tail diameter (from POD 12 to POD 29) Increased the number and suppressed dilation of lymphatic vessels (POD 14) Decreased the number of infiltrated leukocytes in subcutaneous tissue (POD 14) Only a few implanted GFP-labeled ADRC differentiated in LECs (POD 29) Increased plasma VEGF-C level and mRNA expression of *VEGF-C* and *HGF* at surgery site (POD 5) Detected the co-localization of GFP-labeled ADRC and VEGF-C, and LYVE-1 and CD11b/CD163 positive cells (M2 macrophages) Induced M2 macrophages as LEC progenitors
Ackerman et al. 2015 [39]	Male C57BL/6 mouse	Tail model Surgery alone	Mouse inguinal fat pad origin ASC (passage 3, CD31−, CD45−, CD29+, CD90+)	0.5 mL of ASCs/mouse (not described about concentration) + Tegaderm^TM^ dressing Local injection into the wound at POD 0	Vehicle alone Platelet-rich plasma (PRP) prepared from human fresh blood	Decreased wound size in ASC- and PRP-treated group (POD 14) Increased tail volume in ASC-treated group and decreased it in PRP-treated group (POD 7) Increased lymphatic vessel density in PRP-treated group and did not occur in ASC-treated group Increased wound perfusion in ASC- and PRP-treated group (POD 14)
Yoshida et al. 2015 [40]	Male C57BL/6J mouse	Hindlimb model Surgery + RT X-ray irradiation, 30 Gy, 1 week before surgery	Mouse intra-abdominal and -inguinal origin ASC (up to 5 passages)	1.0 × 10^4^, 1.0 × 10^5^, 1.0 × 10^6^ cells/animal Local injection into 5 points at operated limb at POD 2	Vehicle alone	Decreased circumferential length of edematous limb in 1.0 × 10^5^ and 1.0 × 10^6^ ASCs transplantation groups (POD 16) Detected the functional regeneration of collecting lymphatic vessels in 1.0 × 10^6^ ASCs transplantation group and regeneration of capillary lymphatic vessels in 1.0 × 10^4^ and 1.0 × 10^5^ ASCs transplantation groups by lymphography (POD 16) Increased LYVE-1 positive, VEGF-C positive, and VEGFR-3 positive cells dependent on the implanted cell number (POD 16) Few GFP-transgenic mice derived ASCs engrafted in wild-type lymphedema model mice (POD 16) Few male mice-derived ASCs engrafted in female lymphedema model mice confirmed by chromosomal FISH (POD 16, 30)
Hayashida et al. 2017 [41]	Male C57BL6J mouse	Hindlimb model Surgery + RT X-ray irradiation, 30 Gy, 7 days before surgery	Mouse intra-abdominal and -inguinal origin ASC (from 1 to 3 passages)	1.0 × 10^4^ cells/animal + VLNT Local subcutaneous injection in proximal and distal side to the flap at POD 0	Vehicle alone VLNT alone ASC alone	Decreased hind-paw volume in ASC + VLNT group (POD 14) Detected the functional regeneration of collecting lymphatic vessels in ASC + VLNT group by lymphography (POD 14) Increased LYVE-1 positive lymphatic vessels in ASC + VLNT and ASC alone groups (POD 14) Not detected VEGF-C- or VEGFR-3-expressing cells in lymphatic vessels (POD 14) Developed distant metastasis of the B16 melanoma cells from hind paw to trunks skin via transferred lymph nodes in ASC + VLNT group (POD 21)
Beerens et al. 2018 [42]	Female athymic nude Foxn1 mouse	Forelimb model Surgery alone	Human bone marrow origin Multipotent adult progenitor cells (MAPCs)	0.5 × 10^6^ cells human MAPCs in Matrigel + lymph node transfer + Tegaderm dressing Applied into axillary lymph node removed pocket at POD 0	Vehicle in Matrigel + lymph node transfer	Decreased edematous volume (16 weeks after treatment) Increased blood vessels around transferred lymph nodes (16 weeks after treatment) Increased LYVE-1-positive cells around transferred lymph nodes (8 weeks after treatment) Detected Prox1/αSMA-positive, LYVE-1-negative collecting lymphatic vessels around transferred lymph nodes (16 weeks after treatment)
Bucan et al. 2020 [96]	Female C57BL/6 mouse	Hindlimb model Surgery + RT X-ray irradiation, 10 Gy × 2 times, 7 days before and 3 days after surgery	Mouse inguinal fat pad origin SVF (passage 0), ASC (passage 2)	1.0 × 10^6^ cells SVF or ASC/animal Local subcutaneous injection in slightly proximally and distally to the wound gap at POD 7	Vehicle alone	Non-significant edematous volume change between three groups was observed throughout 1–8 weeks after surgery Not improved lymphatic clearance (8–9 weeks after surgery) Decreased areas of lymphatic vessel lumens in ASC treatment group (8 weeks after surgery) Only ~10% or 20% of ASCs expressed CD105 or Sca-1, respectively
Dai et al. 2020 [43]	Female C57BL/6 mouse	Hindlimb model Surgery + RT Irradiated by ^139^Cs, 2.25 Gy × 2 times, 3 days before and 2 weeks after surgery	Mouse origin ASC (fleshly isolated, podoplanin-positive)	2 × 10^6^ cells/animal Local injection into multiple positions within two 1-cm distal areas on the front and back of skin flap at 5 weeks after surgery	Podoplanin-negative ASC Unsorted ASC Vehicle alone	Decreased edematous volume in podoplanin-negative ASC and unsorted ASC transplantation groups, and further decreased in podoplanin-positive ASC transplantation group (from 4 weeks to 10 weeks after treatment) Increased LYVE-1 positive lymphatic vessel density in all ASC treatment groups, and further increased in podoplanin-positive ASC transplantation group (2 weeks after treatment) Detected co-localization of GFP and LYVE-1 positive cells in GFP-labeled podoplanin-positive ASC transplantation group (2 weeks after treatment)
Ogino et al. 2020 [44]	Male C57BL/6J mouse	Hindlimb model Surgery + RT X-ray irradiation, 30 Gy, 7 days before surgery	Mouse origin ASC (commercial item, passages 2–4)	7.5 × 10^5^ cells/animal Local subcutaneous injection into distal and proximal part to the incised wound at POD 1	Surgery + vehicle Surgery + RT + vehicle	Increased the number and area of lymphatic vessel (POD 8) Increased the ratio of proliferating LECs in edematous region (POD 8, 14) Improved fibrosis due to normalized collagen fiber orientation (POD 14) Not increased expression of *VEGF-C* mRNA in skin from edematous region (POD 8, 14)
Nguyen et al. 2022 [45]	Female Sprague-Dawley rat	Hindlimb model Surgery + RT X-ray irradiation, 20 Gy, 7 ± 4 days after surgery	Rat inguinal fat pad origin SVF	3.3 × 10^5^ cells seeded BioBridge × 5/animal SVF-seeded BioBridge was implanted in subcutaneous tunnel made by steel trocar plunger and closed with clip, at 1 month after surgery	Non-treatment	Decreased edematous volume (3 months after treatment) Detected the regeneration of lymphatic vessels toward the contralateral inguinal lymph node and ipsilateral axillary lymph node by lymphography (3 months after treatment)
**(Transplantation of LECs or Tregs)**						
Park et al. 2013 [92]	Male BALB/c mouse	Hindlimb model Surgery + RT Irradiated by electron beam, 1500 cGy × 3 times, 5 days after surgery	Mouse gastrocnemius muscle origin Muscle-derived stem cells (MDSCs), after lymphatic differentiation (Prox-1+, VEGFR-3+, podoplanin+)	1 × 10^7^ cells/animal Local injection at 3 different locations in the hindlimb immediately after irradiation at POD 5	Surgery alone Surgery + RT	Non-significant edematous volume change between cell therapy and Surgery + RT groups was observed Improved in lymphatic flow from distal to proximal part of the body in cell therapy group (POD 56) Increased the LYVE-1 positive lymphatic vessel density in cell therapy group (POD 56)
Kawai et al. 2014 [93]	F344/N rnu/rnu nude rat	Tail model Surgery alone	Human origin LEC (CD31+, podoplanin+, LYVE-1+, Prox-1+)	5 × 10^6^ cells/animal Local injection under the integumentary granulation site at PODs 1, 4, 7, 11, and 14	Unpurified human dermal microvascular endothelial cells (HDMEC) Vehicle alone	Decreased the circumference of tail in LEC transplantation group (from POD 14) and unpurified HDMEC transplantation group (from POD 28) Detected the functional lymphatic regeneration in LEC or unpurified HDMEC transplantation group by lymphography (POD 17) Decreased the epidermal thickness in unpurified HDMEC transplantation group and further in LEC transplantation group (POD 36) Increased the density of podoplanin- or LYVE-1-positive vessels in LEC transplantation group (POD 18, 36) Detected the human podoplanin- or human LYVE-1-positive cells in POD 18, but these cells not detected in POD 36
Gousopoulos et al. 2016 [23]	Female C57BL/6J mouse	Tail model Surgery alone	Mouse origin Treg (expanded by IL-2/anti-IL-2 antibody complex, CD4+, CD25+)	0.8–0.9 × 10^6^ cells/animal Systemically injected into tail vain close to the tail base at 1 week after surgery	Vehicle alone	Decreased the tail volume (1 and 2 weeks after transplantation) Decreased the tissue area covered by lymphatic vessels (2 weeks after transplantation) Significantly reduced the expression of *TGF-β1, TNFα, IL-10* mRNA in edematous region (2 weeks after transplantation) Decreased infiltration of CD45+, CD206+, or CD68+ cells in edematous region (2 weeks after transplantation) Decreased the fibrotic tissue deposition (2 weeks after transplantation) Improved the lymphatic transport capacity (2 weeks after transplantation)

Abbreviations: ADRC, adipose-derived regenerative cell; ASC, adipose-derived stem/stromal cell; α-SMA, alpha smooth muscle actin; BMSC, bone marrow-derived stem/stromal cell; FISH, fluorescence in situ hybridization; GFP, green fluorescent protein; HGF, hepatocyte growth factor; LEC, lymphatic endothelial cell; LYVE-1, lymphatic vessel endothelial hyaluronan receptor-1; Prox-1, prospero homeobox protein 1; RT, radiation therapy; Sca-1, stem cells antigen-1; SVF, stromal vascular fraction; TGF-β1, transforming growth factor beta 1; TNF-α, tumor necrosis factor alpha; Treg, regulatory T cell; VEGF, vascular endothelial growth factor; VLNT, vascularized lymph node transfer.

**Table 2 ijms-23-07614-t002:** Cell-based therapies in patients with lymphedema.

Author	Participants	Cell	Treatment	Groups	Outcome	Side Effects
Hou et al. 2008 [97]	BCRL Undergone a breast cancer surgery and/no radiotherapy 5 years before	Autologous BMSC, collected from iliac crest bone marrow	3–10 × 10^9^ cells/patient 0.5 mL/site intramuscular injection around the axillary, including affected chest wall and part of upper arm	BMSC + custom garment (n = 15) CDT (n = 35)	Reduced the volume of edema in affected arms both in BMSC and CDT groups at 1, 3, and 12 months after treatment, BMSC group showed further reduction in 3 and 12 months after treatment. Reduced the pain score both in BMSC and CDT groups at 1, 3, and 12 months after treatment, BMSC group showed further reduction in 3 and 12 months after treatment.	Not described
Maldonado et al. 2011 [98]	BCRL Patients with unilateral lymphedema secondary to mastectomy and lymphadenectomy with no active cancer in the last 5 years	Autologous bone marrow-derived CD34+ cell, collected from iliac crest bone marrow Initiated by subcutaneous injection of G-CSF for 3 days (300 μg/day)	7–56 × 10^6^ cells/patient 0.5–1 mL/site intramuscular injection around the axillary, including the affected chest wall and part of upper arm, at 30–50 site In CST group, performed during first 4 weeks, then discontinued for following 4 weeks, and then performed again for another 4 weeks	Cell therapy alone (n = 10) CST (n = 10)	Reduced the arm volume in both groups at 4 weeks after treatment Cell therapy group showed significant changes in volume throughout the 12-week follow-up, whereas CST group only showed improvement during periods when the CST was used Improved pain score, sensitivity in the affected limb, and mobility of the affected limb in cell therapy group	Not described
Toyserkani et al. 2017–2021 [99,100,101]	BCRL Recurrence-free disease for a minimum of 1 year, ISL stage I or II	Autologous ADRC, collected from abdomen or thigh adipose tissue (mean percentages of cells surface maker: CD34, 43.1%, CD90, 70.2%; CD31, 19.4%; CD73, 20.5%; CD45, 17.1%; CD235a, 33.1%)	5.37 ± 1.08 × 10^7^ cells/patient (mean ± SD) 5 mL ADRC/patient subcutaneous injection in the axilla at 8 points around the scar 28.1 ± 7.8 mL of fat/patient were injected to release axillary scar tissue about 2 h before ADRC transplantation	Scar release + ADRC + lymphedema management with garments (n = 10) * Non-control group	Non-significant decrease in median lymphedema volume after 4 years of follow-up Improved feelings of arm heaviness and arm tension throughout the follow-up period Improved DASH questionnaire score throughout the follow-up period No changes in LYMQOL score Reduced incidence of cellulitis in five patients who had previously had cellulitis (0.92 ± 1.34 per year → 0.46 ± 0.81 per year, *p* = 0.065) 6 of 10 patients down-staged their lymphedema treatment on their own initiative, 1 of 10 patients upstaged her use of compression sleeve No improved lymph function as mean transit time measured by lymphoscintigraphy at 12 months after treatment	No serious adverse events were found Short-term adverse event related to the liposuction and injections were observed
Ismail et al. 2017 [102]	Primary chronic lower limb lymphedema Primary lymphedema precox or tarda, up to stage III	Autologous BMMNC, collected from iliac crest bone marrow Initiated by subcutaneous injection of G-CSF for 5 days (600 μg/day)	Not described about injected cell numbers Injected in following region: around inguinal lymph nodes, along the lymphatics that accompany superficial venous system of the lower limb, into web space of the foot, and circumferentially in the leg with spacing 3–4 cm in between	BMMNC + compression therapy (n = 20) Compression therapy alone (n = 20)	Decreased VAS score of pain and sense of heaviness in BMMNC group at 1, 3, and 6 months after treatment Increased walking ability and overall patient satisfaction in 70% patients in BMMNC group Decreased circumferential measurements in BMMNC group at 3 and 6 months after treatment Increased CD105-positive vessels in 70% of postintervention specimens	No procedure related adverse effects were observed Biopsy site hematoma developed in two patients
Ehyaeeghodraty et al. 2020 [103]	Primary lower limb lymphedema Grade I or II	Autologous PBMC, collected from antecubital vein blood Initiated by subcutaneous injection of G-CSF for 4 days (300 μg/day)	Two amounts of 9.5 ± 6.8 × 10^8^ PBMCs/patient (mean ± SD), containing 2 × 10^6^ CD34+ cells 1 mL/site subcutaneous injection into 80 marked squares on affected lower limb from below the knee to above the ankle (several hours after cell collection and 3 weeks later)	PBMC + bandages for 3 months (n = 10) Non-control group	Slightly improved QOL in 6 of 10 patients Improved transport index at 6 months after treatment Increased podoplanin-positive lymphatic vessels in one patient Affected limb volume decreased in six patients, not changed in three patients, and slightly increased in one patient at 6 months after treatment In 6 of 10 patients, decreased limb volume 3 months later was not restored to the primary amount despite discontinuing compression therapy	No serious adverse effects (described in conclusion, but no adverse effects described in results or discussion)

Abbreviations: ADRC, adipose-derived regenerative cell; BCRL, breast cancer-related lymphedema; BMMNC, bone marrow-derived mononuclear cell; BMSC, bone marrow-derived stem/stromal cell; CDT, complex decongestive physiotherapy; CST, compression sleeve therapy; DASH, disabilities of the arm, shoulder and hand; G-CSF, granulocyte colony-stimulating factor; ISL, International Society of Lymphology; LYMQOL, Lymphedema Quality-of-Life; PBMC, peripheral blood mononuclear cell; QOL, quality of life; SD, standard deviation; VAS, visual analog scale.

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
