# Peer review of "Emerging Anti-Inflammatory Pharmacotherapy and Cell-Based Therapy for Lymphedema"

_ijms, 2022, doi:10.3390/ijms23147614_

Round 1

Reviewer 1 Report

The manuscript 'Emerging anti-inflammatory pharmacotherapy and cell-based therapy for treating lymphedema' by Ogino et al. reviews the role of anti-inflammatory pharmacotherapy for the treatment of lymphedema. It represents a highly relevant and currently understudied topic and gives a good overview on the topic.

I have only a few minor comments:

1. At the beginning (ll34-35) the authors state 'Primary lymphedema develops due to congenital hypoplasia/dysplasia or dysfunction of lymphatic vessels because of some intrinsic factors': I would recommend not using the term congenital because in the context of a genetic condition it suggests appearance of the symptoms directly after birth, which is not the case for various forms of primary lymphedema such as late onset lymphedema (FoxC2). Therefore, I suggest using the term inherited instead of congenital. In addition, a few more words describing 'some intrinsic factors' would be beneficial for the readers.

2. The authors focusses on 2nd Lymphedema, however, I small paragraph on the potential impact of this treatment on primary lymphedema would improve the manuscript as well.

Author Response

Reply to the Reviewers

Re: ijms-1790270

Title: Emerging anti-inflammatory pharmacotherapy and cell-based therapy for lymphedema

Thank you very much for giving chance to improve our manuscript. According to the reviewers’ comments and suggestions, we have revised our manuscript. Our responses to each reviewer’s comments are described below.

Comments by Reviewer 1 and reply to the comments

Minor comment 1. At the beginning (ll34-35) the authors state 'Primary lymphedema develops due to congenital hypoplasia/dysplasia or dysfunction of lymphatic vessels because of some intrinsic factors': I would recommend not using the term congenital because in the context of a genetic condition it suggests appearance of the symptoms directly after birth, which is not the case for various forms of primary lymphedema such as late onset lymphedema (FoxC2). Therefore, I suggest using the term inherited instead of congenital. In addition, a few more words describing 'some intrinsic factors' would be beneficial for the readers.

Reply to Minor comment 1. The “congenital” was changed to “inherited” in line 34. We also added “such as genetic mutations in the signaling pathway for vascular endothelial growth factor C (VEGF-C)” as an example of intrinsic factors. (line 35–36)

Reference [1]: Grada AA, Phillips TJ. Lymphedema: Pathophysiology and clinical manifestations. J Am Acad Dermatol. 2017;77(6):1009–20. doi: 10.1016/j.jaad.2017.03.022.

Minor comment 2. The authors focusses on 2nd Lymphedema, however, I small paragraph on the potential impact of this treatment on primary lymphedema would improve the manuscript as well.

Reply to Minor comment 2. Thank you for your valuable comment. However, there are few reports regarding the anti-inflammatory pharmacotherapy and cell-based therapy for primary lymphedema. We added the sentences “Additionally, bone marrow-derived mononuclear cells and peripheral blood hematopoietic stem cells could decrease edema volume in primary lower limb lymphedema without serious adverse events [101,102]. Although the molecular mechanism underlying their therapeutic efficacy is unclear at present, cell-based therapy is attractive to establish new therapeutic strategies for primary and secondary lymphedema.” in Discussion (line 388–393)

Reviewer 2 Report

Dear author

Thank you for the submission of your article to our journal.I’ve just read your article and felt some problems as follows;

Major problems

・Your discussion is a little poor. And. You should at least clarify the direction of lymphedema therapy.

・Your title includes anti-inflammatory pharmacotherapy and cell-based therapy. But in you discussion, little discussion has been given to the anti-inflammatory pharmacotherapy. You should discuss the anti-inflammatory pharmacotherapy.

Minor problems

Title

Expression “therapy for treating lymph edema” is strange.

Line 79

Figure should be expressed as Figure 1 in the text or the Figure title should be revised as Figure from Figure 1.

Author Response

Reply to the Reviewers

Re: ijms-1790270

Title: Emerging anti-inflammatory pharmacotherapy and cell-based therapy for lymphedema

Thank you very much for giving chance to improve our manuscript. According to the reviewers’ comments and suggestions, we have revised our manuscript. Our responses to each reviewer’s comments are described below.

Comments by Reviewer 2 and reply to the comments:

Major comment 1. Your discussion is a little poor. And. You should at least clarify the direction of lymphedema therapy.

Major comment 2. Your title includes anti-inflammatory pharmacotherapy and cell-based therapy. But in your discussion, little discussion has been given to the anti-inflammatory pharmacotherapy. You should discuss the anti-inflammatory pharmacotherapy.

Reply to Major comments 1 and 2. Thank you for your valuable comment. We added some information and discussion regarding anti-inflammatory pharmacotherapy in 3rd paragraph of Discussion section as described below.

Doxycycline, selenium, and synbiotic supplements have been reported to improve lymphedema symptoms to some extent. However, their efficacies for decreasing ede-ma volume and improving clinical stage of lymphedema are not remarkable. Further-more, their therapeutic mechanisms for lymphedema have not been elucidated. Com-pared to these agents, the therapeutic mechanisms of immunosuppressive agents such as tacrolimus, IL-4/IL-13 neutralizing antibodies and fingolimod have been elucidated. Although these immunosuppressive agents improve the lymphedema symptoms, the symptoms may return to baseline after the treatment was discontinued. Therefore, pa-tients with lymphedema require lifelong treatment to maintain their QOL. However, long-term treatment with immunosuppressive agents such as fingolimod and IL-4/IL-13 neutralizing antibodies may be a risk factor for infection (e.g., cellulitis) [28]. Further evidences of long-term safety and duration of immunosuppressive therapies are necessary to treat lymphedema in clinical settings. Collectively, we consider that pharmacotherapies with these agents cannot be recommended actively for lymphedema at present.

Minor comment 1. Title: Expression “therapy for treating lymph edema” is strange.

Reply to Minor comment 1. We deleted “treating” from the title.

Minor comment 2. Line 79: Figure should be expressed as Figure 1 in the text or the Figure title should be revised as Figure from Figure 1.

Reply to Minor comment 2. We changed “Figure” to “Figure 1”. (line 80 and 125)

Reviewer 3 Report

This manuscript deals with the therapies for treating lymphedema. The authors pointed out the role of CD4+ T cells in inducing inflammation associated with  lymphedema. Thus, they focus the attention on anti-inflammatory drugs employed to treat lymphedema. Furthermore, they introduced the role of MSC in treating lymphedema, mainly in animal models, and on their direct and indirect effects.

The manuscript is well written, concise and informative. It can be a good starting point for the reader to study this topic. 

Author Response

Thank you for your peer-review and kindly comment.

Reviewer 4 Report

Major:
1. The tacrolimus-FK506BP complex inhibits calcineurin PHOSPHATASE. This affects e.g. trasncription of IL-2, not "reducing the transcriptional activation of IL-2.".  
2. IL-2 mostly affect T cell survival see e.g. 10.1038/ncomms6377 and 10.4049/jimmunol.179.2.950
3. Please consider presenting as a figure a detailed mechanism in which various cytokines promotes fibrosis and lymphoedema in general + highlighting their corresponding blockers.
4. Authors need to introduce the animal model of lymphoedema and comment on its weaknesses and strengths.
5. Was there any clear difference in animal studies when the cell dose was considered? The doses as far as I can see differ easily 100-fold between different studies. 10^7 dose per animal would easily be equivalent of around 1-2x10^10 for human (close to what was administered in study by Hou et al.)
6. Authors failed to include some of the recent studies e.g. https://doi.org/10.3390/biology10090934 https://doi.org/10.1089/lrb.2018.0070 - please make sure that you include all the relevant animal studies and clinical trials

Minor:
1. Line 64. MSCs are not COMMONLY used in treatment. This is an overstatement.
2. What is the novelty of the article in relation to those already published e.g. 10.1007/978-3-030-93039-4_26 ?

Author Response

Reply to the Reviewers

Re: ijms-1790270

Title: Emerging anti-inflammatory pharmacotherapy and cell-based therapy for lymphedema

Thank you very much for giving chance to improve our manuscript. According to the reviewers’ comments and suggestions, we have revised our manuscript. Our responses to each reviewer’s comments are described below.

Comments by Reviewer 4 and replies to the comments:

Thank you for your valuable comments and suggestions. According to your comments, we have revised our manuscript. Our manuscript had already received English proofreading by editage (Cactus Communications K.K., Tokyo, Japan). Our response to your comments is described below.

Major comment 1. The tacrolimus-FK506BP complex inhibits calcineurin PHOSPHATASE. This affects e.g. transcription of IL-2, not "reducing the transcriptional activation of IL-2."

Reply to Major comment 1. We are sorry to our misunderstanding of mechanism of action. We revised the sentence to “the complex inhibits the phosphatase activity of calcineurin, thereby reducing the transcription of IL-2 [88]”. (line 201–202)

Reference [88]: Jørgensen KA, Koefoed-Nielsen PB, Karamperis N. Calcineurin phosphatase activity and immunosuppression. A review on the role of calcineurin phosphatase activity and the immunosuppressive effect of cyclosporin A and tacrolimus. Scand J Immunol. 2003;57(2):93–8. doi: 10.1046/j.1365-3083.2003.01221.x.

Major comment 2. IL-2 mostly affect T cell survival see e.g. 10.1038/ncomms6377 and 10.4049/jimmunol.179.2.950

Reply to Major comment 2. Thank you for your valuable comment. According to your suggestion, we revised the sentence as “CD4+ T cells cannot survive in the presence of tacrolimus because optimal autocrine IL-2 signaling is essential to limit apoptosis of effector CD4+ T cells, and to sustain their transition to and persistence as memory cells [89]”. (line 202–205)

Reference [89]: McKinstry KK, Strutt TM, Bautista B, Zhang W, Kuang Y, Cooper AM, Swain SL. Effector CD4 T-cell transition to memory requires late cognate interactions that induce autocrine IL-2. Nat Commun. 2014;5:5377. doi: 10.1038/ncomms6377.

Major comment 3. Please consider presenting as a figure a detailed mechanism in which various cytokines promotes fibrosis and lymphoedema in general + highlighting their corresponding blockers.

Reply to Major comment 3. We totally agree with your comment that a figure presenting the relationship between the various cytokines and their blockers or MSCs is very useful. However, the detailed mechanisms of actions of agents and cells on lymphedema are not fully understood as described in our manuscript. Only TGF-β inhibitors and anti-IL-4/IL-13 antibodies have been reported to specifically block cytokine, resulting in the suppression of tissue fibrosis. Leukotriene B4 inhibitors could improve lymphatic function histologically, but not decrease edema volume. Furthermore, contribution of leukotriene B4 to pathophysiology of lymphedema is unclear. Thus, we could not present the actions of cytokines and their blockers on figure.

Major comment 4. Authors need to introduce the animal model of lymphoedema and comment on its weaknesses and strengths.

Reply to Major comment 4. According to your suggestion, we added the sentences in Discussion section as follows;

“The rodent tail lymphedema model is commonly used to evaluate the therapeutic strategies for secondary lymphedema because this model closely mimics the progression of human lymphedema, including fibrosis, fat deposition, and infiltration of immune cells [103]” (line 354–357)

“Mouse hindlimb lymphedema model is also used as an animal model of lymphedema. This model is created by a combination of surgery and irradiation, and closely represents the chronic lymphedematous state in human. When lymphedema is induced by surgery alone, the edema might resolve spontaneously. Thus, irradiation is often necessary to create the chronic lymphedema model. However, the dose and timing of irradiation are not standardized among studies, and the degree of edema varies widely among studies [103].” (line 360–366)

“Animal lymphedema models such as rabbit, sheep, dog, pig, and monkey have also been used to study chronic or clinically-relevant lymphedema, but the numbers of these studies are limited [103].” (line 367–369)

Reference [103]: Hsu JF, Yu RP, Stanton EW, Wang J, Wong AK. Current Advancements in Animal Models of Postsurgical Lymphedema: A Systematic Review. Adv Wound Care (New Rochelle). 2022;11(8):399–418. doi: 10.1089/wound.2021.0033.

Major comment 5. Was there any clear difference in animal studies when the cell dose was considered? The doses as far as I can see differ easily 100-fold between different studies. 10^7 dose per animal would easily be equivalent of around 1-2x10^10 for human (close to what was administered in study by Hou et al.)

Reply to Major comment 5. Yoshida et al. reported that the degree of lymphangiogenesis and edema volume reduction depended on the dose of cell (Reference No. 40). In their study, adipose-derived MSCs were locally injected to hindlimb lymphedema mouse model at 1×10^4, 1×10^5, and 1×10^6 cells/animal. Additionally, as described in our Discussion (line 407–409), adipose-derived MSCs are heterogenous cell population and its characteristics may vary widely among researchers. Thus, we consider that there is no consensus regarding the appropriate dose of cells in animal and human studies.

Major comment 6. Authors failed to include some of the recent studies e.g. https://doi.org/10.3390/biology10090934 https://doi.org/10.1089/lrb.2018.0070 - please make sure that you include all the relevant animal studies and clinical trials

Reply to Major comment 6. We have already cited these references you listed in our manuscript (Reference Nos. 28 and 90 in revised manuscript). We further added one recent report by Lee et al as reference No. 83.

Reference [83]: Lee H, Lee B, Kim Y, Min S, Yang E, Lee S. Effects of Sodium Selenite Injection on Serum Metabolic Profiles in Women Diagnosed with Breast Cancer-Related Lymphedema-Secondary Analysis of a Randomized Placebo-Controlled Trial Using Global Metabolomics. Nutrients. 2021;13(9):3253. doi: 10.3390/nu13093253.

This information was added as follows; “In a recent study, intravenous administration of sodium selenite improved lymphoedema and elevated the serum levels of corticosterone, LTB4 dimethylamide (an endogenous LTB4-antagonist), and prostaglandin E3 in breast cancer-related lymphedema patients [83]. Elevated levels of these anti-inflammatory substances may be a factor in therapeutic efficacy of selenium.” (line 179–183)

Minor comment 1. Line 64. MSCs are not COMMONLY used in treatment. This is an overstatement.

Reply to Minor comment 1. We deleted “commonly”. (line 65–66)

Minor comment 2. What is the novelty of the article in relation to those already published e.g. 10.1007/978-3-030-93039-4_26 ?

Reply to Minor comment 2. Most of the previous articles on MSC-based therapy for lymphedema have focused on the regenerative potency of MSC, such as promotion of lymphangiogenesis. However, these articles have not addressed the pathogenesis of lymphedema and therapeutic strategies based on the CD4+ T cell accumulation and inflammatory responses. Thus, in this article, we introduced recent findings on the pathogenesis of lymphedema based on the CD4+ T cell accumulation and inflammatory responses, anti-inflammatory pharmacotherapy and cell-based therapy for lymphedema, and discussed their usefulness for lymphedema treatment in the clinical settings. Furthermore, the characteristics of MSCs (origin, surface markers) and the method of creating lymphedema models are summarized in Table 1. This is the first article to summarize the characteristics of MSCs for treating lymphedema.

Round 2

Reviewer 2 Report

Dear editor

Thank you for the re-submission of your article. I've just read and confirmed the manuscript revised well.

Author Response

Reply to the Reviewers

Re: ijms-1790270

Title: Emerging anti-inflammatory pharmacotherapy and cell-based therapy for lymphedema

Comments by Reviewer 2 and reply to the comments

Thank you for the re-submission of your article. I've just read and confirmed the manuscript revised well.

Reply to the Reviewer 2’s comment: Thank you for your kindly comments.

Reviewer 4 Report

I would still strongly suggest producing the figure as mentioned before. Despite the limited understanding there is a considerable amount of information already accumulated and presenting it as a figure will aid the reader in understanding it. The uncertainties can be then directly mentioned in the figure caption.

Author Response

Reply to the Reviewers

Re: ijms-1790270

Title: Emerging anti-inflammatory pharmacotherapy and cell-based therapy for lymphedema

Comments by Reviewer 4 and replies to the comments:

I would still strongly suggest producing the figure as mentioned before. Despite the limited understanding there is a considerable amount of information already accumulated and presenting it as a figure will aid the reader in understanding it. The uncertainties can be then directly mentioned in the figure caption.

Reply to the Reviewer 4’s comment: We added Figure 2 to illustrate the relationship between some cytokines which promote lymphoedema and pharmacological actions of therapeutic agents for lymphedema (line 143–146)